# Sex Differences in Risk Factors for Metabolic Syndrome in the Korean Population

**DOI:** 10.3390/ijerph17249513

**Published:** 2020-12-18

**Authors:** Yunjeong Yi, Jiyeon An

**Affiliations:** Department of Nursing, Kyungin Women’s University, Incheon 21041, Korea; yinyis@kiwu.ac.kr

**Keywords:** metabolic syndrome, obesity, lifestyle, health behavior, alcohol consumption, socioeconomic status

## Abstract

With an increase in the obese population, the prevalence of metabolic syndrome is increasing in Korea. This study aimed to identify sex- and age-specific risk factors for metabolic syndrome. A secondary data analysis was performed using the Korean National Health and Nutritional Examination Survey. Participants comprised 6144 adults aged 20–79 years. The prevalence of metabolic syndrome was high in the middle- and old-aged men (31.9% and 34.5%, respectively) and in old-aged women (39.1%). Risk factors for metabolic syndrome showed different patterns for men and women. In men, alcohol drinking was identified as the main risk factor for hypertension (odds ratio (OR); young = 3.3 vs. middle age = 2.0), high triglycerides (young = 2.4 vs. middle age = 2.2), and high fasting blood sugar (middle age = 1.6). In women, the main risk factors were household income and education level, showing different patterns in different age groups. In conclusion, the vulnerable groups at high risk of metabolic syndrome are those of middle-aged men and women. The pattern of risk factors is sex-specific.

## 1. Background

Metabolic syndrome is a group of conditions, which together increase the risk of developing atherosclerotic cardiovascular disease, insulin resistance and diabetes mellitus, and vascular and neurological diseases. Metabolic syndrome is associated with high all-cause mortality as well as mortality due to cardiovascular disease [1]. The main causes of metabolic syndrome are abdominal obesity, high blood pressure, diabetes, and dyslipidemia. With an increase in the obese population worldwide, the prevalence of metabolic syndrome is increasing [2,3]. The global prevalence of metabolic syndrome varies slightly depending on the definition of each component and ranges from 24.3% to 45.5% [4,5].

Since metabolic syndrome is a cluster of factors, it is difficult to manage and treat this condition compared to other diseases. The pathogenesis of metabolic syndrome can be described by a complex mechanism. Overweight and obesity are central to the development of metabolic syndrome, predisposing to hypertension, insulin resistance, and dyslipidemia, all of which are risk factors of metabolic syndrome. Physical inactivity and fatty food intake are major causes of obesity [6]. The metabolic abnormality triggered by some intrauterine conditions is accompanied by epigenetic changes [7]. Studies showed that metabolic syndrome is caused by epigenetic modifications rather than genetic predisposition [8], suggesting that external environmental factors lead to changes in gene expression that cause metabolic syndrome.

The external environmental factors associated with metabolic syndrome have been identified as demographic and social characteristics, socio-economic factors, and health behavior and lifestyle [7]. Most countries have national health policies for the management of metabolic syndrome. The public policy for this condition aims to correct lifestyle habits, such as improving eating habits, increasing physical activity, encouraging smoking cessation, regulating alcohol drinking, and managing stress. Despite the implementation of the national health policy for metabolic syndrome, it is difficult to manage the risk factors associated with this condition during adulthood because health behavior and lifestyle are developed through the interaction of genetic, social, and environmental factors since childhood. Moreover, there are several barriers to lifestyle changes due to heterogeneity between different social groups [9]. Individual heterogeneity may be affected by behavioral or environmental effects, but biological sex differences are affected by environmental influences with aging, resulting in more remarkable differences [10]. Therefore, individual heterogeneity, associated with metabolic syndrome development, is primarily explained by sex and age. As people age, they face increasingly complex and often interrelated problems, including physical, psychological, and social problems. Sex differences in metabolic syndrome, in addition to age differences, increase individual heterogeneity. Sex- and age-related factors of metabolic syndrome are sensitive to biological, environmental, and psychosocial conditions [11]. For this reason, sex and age are statistically treated as control variables or used as mediating variables. In previous studies, the hypothesis that the pattern of eating habits directly affects the components of the metabolic syndrome was rejected, and in the end, it was explained that sex differences influenced the linear relationship between the pattern of eating habits and metabolic syndrome [12]. Moreover, sex differences in risk factors for metabolic syndrome are caused by age differences; consequently, sex differences in metabolic syndrome prevalence are due to physiological differences such as hormones, differences in social and psychological stressors, and differences in lifestyle. This is formed integrally [13]. Differences in sex or age appear not only as risk factors for metabolic syndrome but also for mortality from metabolic syndrome.

According to a recent meta-analysis on metabolic syndrome, low high-density lipoprotein cholesterol (HDL-C), hyperglycemia, and elevated blood pressure have higher association with all-cause mortality and cardiovascular mortality in women [14]. Differences in sex and age in the incidence of and mortality from metabolic syndrome can be explained by differences in risk factors. As Korea quickly transitions from being a developing to a developed country, studies on obesity and metabolic syndrome are increasing due to westernized dietary habits and increased sedentary lifestyle. With ongoing research on the risk factors for metabolic syndrome, accumulated knowledge on metabolic syndrome risk factors is still a necessity. Most previous studies only investigated differences according to sex or restricted their investigation to the elderly population only [15,16]. As such, when determining the risk factors for metabolic syndrome, it is necessary to comprehensively consider the influence of sex and age; thus, national policies for metabolic syndrome management may be more specific. Herein, we aimed to identify the sex- and age-specific risk factors for metabolic syndrome in the Korean population and attempted to determine whether any pattern exists among the risk factors.

## 2. Materials and Methods

### 2.1. Research Design

This study used the 2017 survey data of the Korea National Health and Nutrition Examination Survey (KNHANES). It was a secondary data analysis study to determine how the factors influencing each variable of metabolic syndrome vary with sex and age.

### 2.2. Setting

The KNHANES has been conducted since 1998, and the sampling frame of the survey used the most recent census data and public prices of apartments available at the point of sampling. The sampling method was a two-stage stratified cluster sampling method in which the primary extraction unit was a survey unit and the secondary extraction unit a household. In the 2017 survey, 192 households were surveyed, and 23 sample households were selected using the system extraction method. All subjects aged 1 year and above were selected for the survey.

### 2.3. Participants

This study used the 2017 survey data with 3580 households participating and 8127 participants (proportion of men, 46.3%). In total, 6144 people were analyzed, with age between 20 and 79 years, and the proportion of men was 45.1%.

### 2.4. Variables

#### 2.4.1. Dependent Variable

The dependent variables in this study were the five variables used in diagnosing metabolic syndrome: waist circumference, blood pressure, triglycerides, fasting blood sugar, and HDL-C. These standards were announced by the American Heart Association/National Heart, Lung, and Blood Institute Scientific Statement [17].

This clinical definition of metabolic syndrome required the presence of three or more of the following five criteria: (1) waist circumference over 90 cm for men and over 85 cm for women; (2) blood pressure above 130 mmHg for systolic or 85 mmHg for diastolic blood pressure; (3) triglyceride levels of 150 mg/dL or higher; (4) fasting blood sugar level of 100 mg/dL or higher; (5) HDL-cholesterol level less than 40 mg/dL for men and less than 50 mg/dL for women.

#### 2.4.2. Independent Variables

The independent variables included sex, age, household income, education level, subjective health status, monthly alcohol intake, stress recognition, smoking status, aerobic physical activity, number of “eating out” episodes per day, and living with spouse. The subjects were divided into young (20–39 years old), middle-aged (40–59 years old), and old-aged (60–79 years old), and income was divided into quintiles using the household income.

The level of education was classified into “below high school” and “above college.” Regarding the subjective health status, the five-point scale from “very bad” to “very good” was divided into two categories: “under average” and “over good.” Drinking status was categorized as “drunk more than once a month in the past year” or “drunk once per month”.

Stress recognition was divided into two categories: “a lot” and “little” based on an unusual four-point scale of “how much stress you usually feel in your daily life?”.

A smoker was defined as a person who smoked more than five packs (100 cigarettes) in his lifetime and at the same time, smokes daily. Aerobic physical activity was considered if a person engaged in more than 2.5 h of high-intensity physical activity per week, more than 1.25 h of moderate-intensity physical activity, or the time equivalent to each activity by mixing moderate- and high-intensity physical activity (1 min of high-intensity equates to 2 min of moderate-intensity physical activity). Based on dining out, people who eat out more than once a day and those who do not eat out were distinguished.

The analysis criteria of the KNHANES were used to categorize subjective health status, monthly alcohol intake, stress recognition, smoking, aerobic physical activity, and dining out. These criteria are used by the Ministry of Health and Welfare in its “annual statistics book” based on data from the KNHANES.

### 2.5. Data Measurement

The KNHANES collects data through household member confirmation surveys, health questionnaire surveys, examination surveys, and nutrition surveys. Health questionnaire surveys were divided into household surveys, health interview surveys, and health behavior surveys. The family survey was conducted in the form of an interview survey on one adult (over 19 years of age) in the household. The health interview survey investigated the level of education and physical activities by the interview survey method, whereas the health behavior survey investigated smoking, drinking, mental health, etc., by the self-reported survey method. The examination surveys consisted of physical measurement, blood pressure and pulse measurement, blood and urine test, oral examination, lung function test, eye test, and grip test.

### 2.6. Statistical Methods

Since the KNHANES data were from two-stage stratified cluster sampling rather than a full-scale survey, they were analyzed to reflect the complex sampling elements, which can be expanded to the target group, the Korean population. A complex sample analysis with strata, cluster, and weights was performed, and the missing values were valid, and the sex and age-group data were applied to the parent group. In frequency analysis, non-weighted frequencies were calculated as statistics, and odds ratios (ORs) were calculated through complex sample logistic regression.

### 2.7. Ethical Considerations

The KNHANES was performed without being reviewed by the Research Ethics Review Committee from 2015 under the provisions of the Bioethics Act, which falls under the research conducted directly by the state for public welfare. The raw data for this study were downloaded and analyzed after signing and submitting the “Statistics User’s Compliance Statement” and “Security pledge” on the website of the National Health and Nutrition Survey.

## 3. Results

The general characteristics of the study subjects according to sex and age are shown in Table 1. The old-aged group had a lower income and a lower education level than that of the other age groups. The young age group had a higher monthly alcohol intake, stress recognition, aerobic physical activity, and number of instances of “dining out” per day than that of other age groups. Subjective health and smoking habits differed in the dominant age groups by sex (Table 1).

The prevalence of each variable of metabolic syndrome in men and women of different age groups is shown in Table 2. In women, the prevalence of all variables was highest in the old-aged group (*p* < 0.001), whereas in men, the prevalence of hypertension and high triglyceride level (*p* < 0.001) was highest in the middle-aged group; no difference in waist circumference was observed between different age groups (Table 2). In men, the prevalence of metabolic syndrome was more than 30% in middle-aged and old-aged groups (*p* < 0.001), whereas in women, the prevalence was more than 30% in the middle-aged group only (*p* < 0.001).

The factors influencing each component of the metabolic syndrome in males and in females of different age groups are presented in Table 3, Table 4 and Table 5. In men, subjective health status and monthly alcohol drinking were the main variables of all ages (Table 3). Alcohol drinking was a risk factor for hypertension (OR = 2.0, *p* < 0.001), high triglyceride levels (OR = 2.2, *p* < 0.001), and increased fasting blood sugar (OR = 1.6, *p* < 0.05) in middle-aged men.

More variables affected women than men and included household income, education level, monthly alcohol drinking, and subjective health status (Table 4). Factors affecting each sex and different age groups were very different. Household income was identified as a prominent risk factor in women. The middle-aged group had the highest number of risk factors in both men and women. In males, the young age group was mainly affected by monthly alcohol drinking, the middle-aged group was affected by monthly alcohol drinking and subjective health status, and the old-aged group was affected by subjective health and smoking. In women, the middle-aged group was mainly affected by education levels and subjective health status, whereas the old-aged group was mainly affected by income levels. In both men and women, most of the variables affected the middle-aged group. In all age groups, women had more influencing variables than men (Table 5).

## 4. Discussion

This study investigated risk factors for metabolic syndrome by considering sex and age differences. Unlike previous studies, the study subjects included both adults and the elderly. Regarding the prevalence rate of each component of metabolic syndrome, there were differences in frequency by sex and age, excluding waist circumference. Compared to men, women generally had a higher prevalence rate for each component of metabolic syndrome in the elderly than in young adults. These results support the existing research findings that women experience physiological changes that make them susceptible to metabolic syndrome with aging. Menopause, characterized by a decline in circulating estrogen levels, may increase cardiovascular risk through its effects on adiposity and lipid metabolism [18]. In fact, it is known that postmenopausal women have more than double the prevalence of metabolic syndrome compared to premenopausal women, especially with low HDL-C, high blood pressure, and high fast glucose levels [19].

The implications of the risk factors for metabolic syndrome by sex and age are as follows. The middle-aged people, regardless of sex, are exposed to more risk factors associated with metabolic syndrome than any other age group. Thus, the target risk group for managing metabolic syndrome is that of middle-aged people for both men and women. Compared to other age groups, middle-aged people spend a lot of time on economic activities; therefore, they neglect their own health care.

The risk factors for metabolic syndrome differed by sex. In middle-aged men, alcohol consumption was a risk factor for hypertension and high triglycerides, and in middle-aged women, education level was a risk factor for all five components of metabolic syndrome. The differences between men and women in risk factors related to metabolic syndrome may be due to differences in health behavior and lifestyle of men and women.

In general, excessive energy intake and concomitant obesity cause metabolic syndrome, resulting in atherosclerotic cardiovascular disorder and type 2 diabetes [20]. The main mechanism by which these complications arise from metabolic syndrome can be explained by vascular thickness and stiffness [21]. Among the pathological changes in blood vessels, vascular stiffness causes arteriosclerosis, and the blood vessels are narrowed due to fatty deposits. Thus, obesity-related diet and lifestyle are directly related to metabolic syndrome [22]. Among the obesity-related eating habits is alcohol consumption. Previous studies have identified that alcohol consumption cause metabolic syndrome because alcohol stimulates food consumption, resulting in an increase in total energy consumption [23]. In general, because the frequency and amount of alcohol intake is greater in men than in women, the negative effects of alcohol are more pronounced in men [24].

The association between the quantity of alcohol intake and metabolic syndrome is debatable. According to a previous study, considering both the quantity and the frequency of drinking, heavy alcohol consumption increases the risk of metabolic syndrome, while light alcohol consumption lowers the risk of metabolic syndrome [25,26]. Furthermore, it is reported that the link between drinking and metabolic syndrome is more pronounced in men than in women [27]. We observed that in men, alcohol consumption is a risk factor for metabolic syndrome, with relatively high OR values in young and middle-aged people. While alcohol consumption was identified as a risk factor in men, it was identified as a protective factor in women. This can be explained based on differences in the quantity and frequency of alcohol intake between men and women or due to interaction with other variables.

In middle-aged women, education level was identified as a risk factor in all five categories of metabolic syndrome. In women (young, middle-aged, and old) household income was shown as a risk factor for certain components of metabolic syndrome. Socioeconomic status, such as education level and household income, was reported to be a major risk factor for metabolic syndrome, especially in women.

The close association between socioeconomic status and metabolic syndrome, which is usually prominent in women, could be explained by the stress theory. In females, chronic psychosocial stress induces changes in the hypothalamus-pituitary-adrenal axis, resulting in increased insulin resistance or adipocyte proliferation, which ultimately causes metabolic syndrome. Stress influences health risk behaviors (smoking, low physical activity, drinking, and eating fat and sugar-rich products) leading to the development of metabolic syndrome [28]. Few studies suggest that low economic status is a specific barrier to change in lifestyle. Sparing time for physical activity or ensuring a budget to buy healthier food is not easy in the low-income group. Moreover, low-fat food or slow food is too expensive, and it is costly to use fitness centers to support lifestyle change [9].

In Korean women, education level and household income are associated with the risk of metabolic syndrome. With a low education level or low household income, they are likely to do domestic work as well as jobs for livelihood, which is a scenario known to cause metabolic syndrome due to increased stress and economic deprivation [29]. The study of the association between socioeconomic status (SES) and metabolic syndrome uses subjective SES. Subjective SES is known to increase the risk of central adiposity (waist-hip ratio), low HDL cholesterol, elevated serum triglycerides, and blood pressure in middle-aged women [30].

Our results are consistent between men and women. Physical activity has been shown to affect the waist circumference for both middle-aged men and women. The linear relationship between physical activity and obesity is already proven. According to a study conducted on twins, physical activity reduces the risk of developing high BMI and high waist circumference [31].

Obesity is a health issue that is most prominent in middle-aged people. A prospective study involving young adults aged 18–30 years who were not initially obese showed that more than 40%of these individuals developed overall and abdominal obesity during middle age and obesity was linked to the occurrence of subclinical heart disease [32]. It is the fact that obesity in adults lasts for a long time, which consequently leads to the development of atherosclerosis during middle age. Some researchers showed that the risk of metabolic syndrome increases after middle age because metabolic syndrome is affected by the aging process [33]. With aging, the risk of metabolic syndrome as well as the risk of cardiovascular disease increases. The management of metabolic syndrome in people of all age groups is necessary. However, in the middle-aged group, since the aging process is accelerated during this period, the management of metabolic syndrome needs to be focused [34]. Abdominal obesity management in middle-aged men and women is important healthcare that can prevent cardiovascular disease as well as metabolic syndrome.

In this study, we also investigated the association between the external environmental factors on different components of the metabolic syndrome in men and women of different age groups. Drinking is a risk factor in men; however, in women, it appears to be a protective factor for some components of metabolic syndrome. Regarding smoking, it was a protective factor for waist circumference; however, for dyslipidemia, it was found to be a risk factor. Smoking is known to be associated with low HDL-C and high triglyceride as well as with pathological changes in the blood vessels [35]. This study also showed that smoking was associated with high triglyceride and low HDL-C levels in older men, which is consistent with previous studies.

In summary, this study found that the prevalence of metabolic syndrome differs by sex and age and shows that the risk factors for metabolic syndrome that can be intervened also differ by sex and age. Although it is difficult to determine how risk factors cause metabolic changes in each sex and age group, this study suggests that the management of metabolic syndrome needs to integrate genetic and acquired influencing factors. Metabolic syndrome requires integrated treatment due to comorbidities. It is also a chronic and complex disease that increases the risk of cardiovascular disease and type 2 diabetes and is associated with high mortality. Since some individuals are predisposed to risk factors and the risk of complications associated with metabolic syndrome, some people require lifestyle modification, while others need direct pharmacological treatment [7]. Therefore, screening is required for high-risk groups, and customized guidelines are needed.

There are several limitations to our study. Although our study included a large sample, it does not include dietary-related variables related to metabolic syndrome and used some variables from the self-reporting survey. Unlike previous studies, our study analyzed factors affecting metabolic syndrome according to sex and age and identified patterns of related variables.

## 5. Conclusions

In summary, the target risk groups for metabolic syndrome are middle-aged men and women, and the risk factors for men and women are different. Middle-aged men are advised not to drink and smoke excessively, and women need to minimize unmet healthcare needs due to economic stress through preventive health service programs. For women, the risk factors of metabolic syndrome were higher than those in men. Women become more susceptible to metabolic syndrome with aging; hence, intensive interventions are needed after middle age to address the risk factors. Thus, different guidelines are required for men and women. These results will be applicable not only to the Korean population but also to people who have unhealthy health behaviors while living a competitive life globally. In particular, in all the countries with an increasing obese population, increasing interest in sex- and age-related risk factors for metabolic syndrome is required.

## Figures and Tables

**Table 1 ijerph-17-09513-t001:** General characteristics of subjects by sex and age groups (*n*, %).

		Male				Female			
		20–39 (*n* = 789)	40–59 (*n* = 1131)	60–79 (*n* = 852)	Chi-Square (*p*)	20–39 (*n* = 918)	40–59 (*n* = 1352)	60–79 (*n* = 1102)	Chi-Square (*p*)
Household income	Low	42 (6.7)	76 (5.9)	199 (22.0)	182.5 (<0.001)	29 (3.2)	81 (5.7)	370 (34.7)	328.7 (<0.001)
	Middle low	90 (12.5)	135 (12.4)	220 (26.1)		130 (14.7)	197 (15.2)	280 (25.1)	
	Middle	180 (25.5)	192 (19.5)	185 (23.5)		222 (27.5)	282 (20.9)	192 (17.8)	
	Middle high	206 (29.0)	284 (27.0)	113 (14.6)		253 (28.8)	332 (26.6)	117 (11.9)	
	High	207 (26.3)	362 (35.2)	106 (13.9)		221 (25.8)	408 (31.7)	107 (10.5)	
Education Level	Under high school	281 (42.4)	489 (48.9)	615 (77.4)	102.1 (<0.001)	252 (29.8)	737 (59.9)	938 (92.2)	282.1 (<0.001)
	Over college	411 (57.6)	483 (51.1)	159 (22.6)		566 (70.2)	499 (40.1)	73 (7.8)	
Subjective health status	Over good	251 (36.3)	273 (27.9)	222 (29.5)	11.9 (0.003)	255 (29.9)	338 (26.2)	136 (13.0)	64.4 (<0.001)
	Under average	443 (63.7)	706 (72.1)	560 (70.5)		565 (70.1)	904 (73.8)	879 (87.0)	
Monthly alcohol drinking	<1	169 (23.6)	251 (24.5)	297 (34.3)	22.5 (<0.001)	329 (38.0)	695 (53.1)	812 (76.9)	213.4 (<0.001)
	≥1	554 (76.4)	786 (75.5)	521 (65.7)		520 (62.0)	596 (46.9)	246 (23.1)	
Stress recognition	No	458 (64.1)	763 (73.6)	685 (83.4)	60.1 (<0.001)	534 (61.2)	938 (72.4)	786 (73.9)	37.3 (<0.001)
	Yes	265 (35.9)	274 (26.4)	132 (16.6)		315 (38.8)	351 (27.6)	267 (26.1)	
Smoking	No	433 (59.5)	605 (58.3)	621 (76.2)	54.4 (<0.001)	785 (92.1)	1237 (95.5)	1030 (97.5)	27.2 (<0.001)
	Yes	290 (40.5)	432 (41.7)	195 (23.8)		64 (7.9)	52 (4.5)	26 (2.5)	
Aerobic physical activity	No	290 (41.4)	516 (52.9)	510 (64.1)	48.8 (<0.001)	403 (47.2)	674 (55.2)	698 (68.3)	65.2 (<0.001)
	Yes	401 (58.6)	457 (47.1)	263 (35.9)		415 (52.8)	562 (44.8)	308 (31.7)	
Number of instances of dining out per day	<1	292 (49.6)	503 (58.0)	674 (88.1)	160.6 (<0.001)	557 (74.7)	1005 (85.8)	935 (96.0)	106.1 (<0.001)
	≥1	295 (50.4)	333 (42.0)	79 (11.9)		187 (25.3)	173 (14.2)	38 (4.0)	
Living with spouse	Yes	291 (98.7)	859 (92.1)	728 (90.5)	14.7 (0.001)	467 (96.8)	1104 (88.5)	671 (63.2)	209.0 (<0.001)
	No	5 (1.3)	80 (7.9)	84 (9.5)		17 (3.2)	151 (11.5)	390 (36.8)	

**Table 2 ijerph-17-09513-t002:** Prevalence of metabolic syndrome components according to sex and age groups (*n*, %).

		Male				Female			
		20–39 (*n* = 789)	40–59 (*n* = 1131)	60–79 (*n* = 852)	Chi-Square (*p*)	20–39 (*n* = 918)	40–59 (*n* = 1352)	60–79 (*n* = 1102)	Chi-Square (*p*)
Waist circumstance	No	482 (68.1)	674 (67.3)	491 (62.8)	4.6 (0.101)	739 (88.4)	991 (78.4)	578 (57.6)	133.4 (<0.001)
	Yes	225 (31.9)	329 (32.7)	294 (37.2)		104 (11.6)	279 (21.4)	441 (42.4)	
Hypertension	No	532 (73.1)	632 (60.0)	494 (61.4)	25.7 (<0.001)	81.2 (94.7)	998 (78.0)	558 (54.2)	241.9 (<0.001)
	Yes	195 (26.9)	420 (40.0)	330 (38.6)		48 (5.3)	303 (22.0)	511 (45.8)	
Triglyceride	No	456 (65.1)	556 (54.2)	515 (64.8)	25.2 (<0.001)	735 (89.5)	1007 (79.5)	711 (71.2)	61.7 (<0.001)
	Yes	254 (34.9)	472 (45.8)	283 (35.2)		93 (10.5)	257 (20.5)	292 (28.8)	
Fasting blood sugar	No	562 (80.8)	520 (50.9)	346 (44.2)	215.9 (<0.001)	760 (92.5)	913 (72.4)	515 (51.8)	237.4 (<0.001)
	Yes	148 (19.2)	508 (49.1)	452 (55.8)		67 (7.5)	351 (27.6)	488 (48.2)	
HDL cholesterol	No	570 (82.1)	752 (73.2)	548 (70.3)	24.9 (<0.001)	612 (75.4)	837 (67.1)	506 (51.0)	88.0 (<0.001)
	Yes	130 (17.9)	267 (26.8)	238 (29.7)		207 (24.6)	407 (32.9)	487 (49.0)	
Metabolic syndrome	No	547 (80.9)	662 (68.1)	489 (65.5)	43.4 (<0.001)	765 (95.6)	999 (82.9)	563 (60.9)	156.6 (<0.001)
	Yes	134 (19.1)	310 (31.9)	258 (34.5)		39 (4.4)	215 (17.1)	384 (39.1)	

**Table 3 ijerph-17-09513-t003:** Factors affecting metabolic syndrome components in different male age groups.

Variables	Reference	Categories	Odds Ratios in Male Subjects																	
			Waist Circumstance			Hypertension			Triglyceride			Fasting Blood Sugar			HDL Cholesterol			Metabolic Syndrome		
			20–39	40–59	60–79	20–39	40–59	60–79	20–39	40–59	60–79	20–39	40–59	60–79	20–39	40–59	60–79	20–39	40–59	60–79
Household income	High	Low	3.5	0.9	1.3	0.4	1.3	1.3	4.0	2.7 *	1.0	0.1	1.4	1.4	0.0 **	1.5	1.3	0.5	1.9	2.0
		Middle low	1.1	1.0	1.0	0.5	0.6	1.1	0.9	1.6	1.2	0.5	1.5	1.0	0.8	1.2	1.1	0.4	1.7	1.3
		Middle	0.8	1.2	1.0	0.7	0.9	1.2	1.0	1.0	1.0	0.5	0.8	1.0	1.3	1.1	1.3	0.5	1.4	1.2
		Middle high	0.5	1.1	1.3	0.8	0.9	1.2	1.3	1.0	1.3	0.4 *	1.3	0.8	2.1	1.0	1.0	0.4	1.5	1.5
Education level	Over college	Under high school	1.0	1.4	1.0	0.6	1.2	0.7	1.7	0.8	0.9	0.9	1.3	2.0 **	0.9	0.8	0.8	1.3	0.9	1.2
Subjective health status	Over good	Under average	1.6	1.3	1.3	1.4	1.6 *	1.3	1.4	1.5 *	1.7 *	1.4	1.4	1.5 *	2.4	1.1	1.1	1.4	1.7 *	1.9 *
Monthly alcohol drinking	<1	≥1	1.3	1.1	0.8	3.3 *	2.0 **	1.4	2.4 *	2.2 **	1.0	1.1	1.6 *	1.4	0.6	0.7	0.4 **	2.2	1.9 *	0.9
Stress recognition	No	Yes	0.6	1.1	0.7	0.8	1.0	0.7	1.4	1.1	1.1	1.1	1.3	1.1	0.4 *	1.0	1.0	0.7	1.0	0.7
Smoking	No	Yes	1.0	0.7 *	1.1	1.4	0.7 *	1.1	1.4	1.4	1.6 *	1.2	1.0	0.9	1.6	1.3	1.9 **	1.7	1.0	1.2
Aerobic physical activity	No	Yes	0.6	0.7 *	1.3	1.2	1.1	1.1	1.0	0.9	0.9	1.4	0.7	1.4	1.2	0.7	0.7	0.9	0.7	0.9
Number of instances of dining out per day	<1	≥1	1.0	0.9	1.3	1.0	0.9	1.0	1.1	0.9	0.9	1.1	0.6 **	2.0 *	0.9	0.7	0.8	1.0	0.6 *	1.2
Living with spouse	Yes	No	3.6	0.7	1.0	2.5	1.1	0.7	1.1	0.7	1.0	1.3	1.0	0.8	2.3	1.3	1.9	1.2	0.5 *	1.1

* *p* < 0.05; ** *p* < 0.01.

**Table 4 ijerph-17-09513-t004:** Factors affecting metabolic syndrome components in different female age groups.

Variables	Reference	Categories	Odds Ratios in Female Subjects																	
			Waist Circumstance			Hypertension			Triglyceride			Fasting Blood Sugar			HDL Cholesterol			Metabolic Syndrome		
			20–39	40–59	60–79	20–39	40–59	60–79	20–39	40–59	60–79	20–39	40–59	60–79	20–39	40–59	60–79	20–39	40–59	60–79
Household income	High	Low	4.3	1.1	1.9 *	0.0 **	2.0	1.1	1.4	1.1	2.3 **	0.7	1.1	1.6	1.6	0.9	2.2 *	1.7	1.0	2.1 *
		Middle low	4.4 *	2.1 *	1.3	1.2	1.2	1.2	1.1	1.3	1.7	2.5	0.8	1.2	1.1	1.2	1.7	1.5	1.6	1.8
		Middle	3.4	1.4	1.4	1.3	1.1	1.2	1.2	0.9	1.0	1.1	0.6 *	1.0	1.8	1.0	1.5	1.7	1.0	1.4
		Middle high	2.8	1.4	1.0	0.9	1.4	0.5 *	1.9	0.7	1.1	1.0	0.8	1.2	2.1 *	0.9	1.5	2.5	0.9	0.8
Education level	Over college	Under high school	1.7	2.0 **	2.0	1.1	1.8 **	1.0	1.4	1.5 *	1.4	1.8	1.7 **	1.4	1.4	1.4 *	1.4	2.5	1.8 **	1.6
Subjective health status	Over good	Under average	1.2	1.7 *	1.5	1.0	1.9 **	1.1	2.1	1.8 *	1.0	0.7	1.3	1.1	1.1	1.2	1.0	1.8	2.2 **	1.2
Monthly alcohol Drinking	<1	≥1	0.5	0.9	0.9	3.3 *	1.1	1.0	1.0	1.1	0.9	1.1	1.0	0.8	0.5 *	0.7 *	0.7 *	0.7	0.8	0.7 *
Stress recognition	No	Yes	1.1	1.4	0.9	0.9	1.2	0.7 *	0.8	1.0	1.1	0.8	0.8	0.9	1.0	1.1	0.8	1.5	1.3	0.8
Smoking	No	Yes	1.7	1.0	0.2 *	1.0	1.1	1.4	1.4	2.0	0.3	2.9	0.5	1.1	4.0 **	1.5	0.9	1.7	1.4	0.8
Aerobic physical activity	No	Yes	1.5	0.6 **	0.9	1.4	1.0	0.9	0.9	0.7 *	0.8	1.8	0.9	0.8	1.0	0.9	0.9	1.6	0.8	0.9
Number of instances of dining out per day	<1	≥1	0.8	1.0	0.5	0.7	0.4 **	1.3	0.7	0.8	0.6	0.6	1.0	1.5	0.9	1.0	0.6	0.3	0.8	0.7
Living with spouse	Yes	No	1.0	1.1	1.3	15.1 **	1.1	1.2	1.6	0.7	0.8	4.3 *	0.9	1.5 *	0.5	0.7	0.7 *	3.4	1.0	0.8

* *p* < 0.05; ** *p* < 0.01.

**Table 5 ijerph-17-09513-t005:** The number of significant risk factors associated with different components of metabolic syndrome in the different age groups of both sexes.

Variables	Male															Female														
	20–39					40–59					60–79					20–39					40–59					60–79				
	WC	HT	TG	FBS	HDL-C	WC	HT	TG	FBS	HDL-C	WC	HT	TG	FBS	HDL-C	WC	HT	TG	FBS	HDL-C	WC	HT	TG	FBS	HDL-C	WC	HT	TG	FBS	HDL-C
Household income					-			+								+	-			+	+			-		+	-	+		+
Education level				-										+							+	+	+	+	+					
Subjective health status							+	+					+	+							+	+	+							
Monthly alcohol drinking		+	+				+	+	+						-		+			-					-					-
Stress recognition					-																						-			
Smoking						-	-						+		+					+						-				
Aerobic physical activity						-															-		-							
Number of instances of dining out per day									-					+								-								
Living with spouse																	+		+										+	-
Number of significant factors	0	1	1	1	2	2	3	3	2	0	0	0	2	3	2	1	3	0	1	3	4	3	3	2	2	2	2	1	1	3

WC: Waist Circumstance; HT: Hypertension; TG: Triglyceride; FBS: Fasting Blood Sugar; HDL-C: HDL-Cholesterol. (-) protective factor; (+) risk factor.

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
