# Peer review of "Sex Differences in Risk Factors for Metabolic Syndrome in the Korean Population"

_ijerph, 2020, doi:10.3390/ijerph17249513_

Round 1

Reviewer 1 Report

Development of the article:

The study is interesting, however, it must be more specific and precise in the bibliography that shows the approach presented is the most interesting and in the methodology several areas for improvement have been found, which are specified below.

Authors are advised to perform a search in PubMed or Web of Science on the reviews published on this topic, with the following strategy: Metabolic syndrome and sex or gender and age, they will find an article like this: Gender differences in the impact of metabolic syndrome components on mortality in older people: A systematic review and meta-analysis. Nutr Metab Cardiovasc Dis. 2020 Aug 28; 30 (9): 1452-1464. doi: 10.1016 / j.numecd.2020.04.034. This recommendation is justified in the introduction section.

In the introduction:

The focus of the objective is based on gender and age, however, the introduction only presents a bibliographic citation in that sense, it is recommended to carry out a bibliographic search on this subject or change the objective of the article. It seems as if initially the authors had an objective and finally this objective changed, therefore it is recommended to carry out a more precise approach depending on the objective that the authors pursue (from lines 53 to 58).

Methodology:

It is not described which study it is, it is understood that it is an observational study, for future studies here it has different protocols to carry out this type of study (http://www.equator-network.org/).

It should indicate the difference between this study and previous published studies, such as: Gender-Specific Associations between Socioeconomic Status and Psychological Factors and Metabolic Syndrome in the Korean Population: Findings from the 2013 Korean National Health and Nutrition Examination Survey. Biomed Res Int. 2016; 2016: 3973197. doi: 10.1155 / 2016/3973197.

The authors obtain all the information on the Korea National Health and Nutrition Examination Survey (KNHANES), however, this tool is not provided, it does not describe what factors this survey records and how it was designed (from lines 66 to 67). One of the most relevant aspects is to visualize in which populations or places this survey was registered and that must be described in the methodology of this tool.

It is indicated that the data from the survey of a second year of the 7th phase (from line 76 to 78) was used, these data are confusing, the authors must describe and justify this decision.

The variables analyzed are identified based on the America National Cholesterol Education Program in 2005 (line 84), but this information has no bibliographic reference.

The clinical definition of metabolic syndrome (line 86) also has no bibliographic citation.

In the independent variables, perform subgroups of studies (from lines 98 to 102), it is recommended that reference be sought from other studies to standardize these concepts or justify whether they are consolidated concepts within the type of survey from previous years.

Regarding the concepts of physical activity intake, frequency of tobacco consumption, it would be advisable to standardize these concepts (from lines 103 to 111), for example, the American Heart Association recommends using certain limits in some of these concepts ( Steinberger J, Daniels SR, Hagberg N, et al. Cardiovascular Health Promotion in Children: Challenges and Opportunities for 2020 and Beyond: A Scientific Statement From the American Heart Association. Circulation. 2016; 134 (12): e236 – e255). The use of these standardized and validated concepts will determine that changes should be made in Table 1, 3 and 4.

When you make the recommended indications and changes then it will be interesting to evaluate the results and the discussion.

Author Response

Thank you for the detailed review.

All revised according to the opinion of reviewer. Attach the revised manuscript.

Reviewer 2 Report

Metabolic syndrome is the medical term for a combination of diabetes, high blood pressure (hypertension) and obesity. It puts you at greater risk of getting coronary heart disease, stroke and other conditions that affect the blood vessels.
Approximately 20–25 percent of the world's adult population has the cluster of risk factors that is metabolic syndrome therefore efforts for prevention, treatment and understanding of the risk factors and the underlying causes of this syndrome are essential.
Therefore, the current manuscript which inctroduce gender- and age-specific risk factors for metabolic syndrome can complement the existing knowledge.

In my opinion the manuscript should be accepted for publication. I only suggest that the Authors should include information on the potential impact of their research on the general population, not only in Korea.

Author Response

(The authors gave the same response as above.)

Reviewer 3 Report

"Sex Differences in Risk Factors for Metabolic Syndrome in the Korean Population" manuscript by Yunjeong Lee , Jiyeon An describes the major risk factors for metabolic syndrome in Korean population giving special attention to the differences between men and women and age differences. In the perspective of this reviewer the study is well presented, the population analyzed is well balanced between the different age and gender groups. The results are also well presented in the form of tables and statistic is well applied. Discussion could be more exhaustive but after a search in the literature the reviewer understand that the studies to compare with are not so abundant. However, the reviewer only have doubts on the significante and soundness of the study for the scientific community since the conclusions are common sense.

The present manuscript aims to discuss the impact of social, behavior and comorbidities risk factors on the prevalence of metabolic syndrome in the Korean population. The authors also discriminate between the differential impact of those risk factors between gender and age groups. The reviewer acknowledges the effort of the authors to include a relevant number of
people in each group and a correct description of it. The results are quite well described, and the major conclusions fits with those results. However, the reviewer would like to suggest that the discussion should be more detailed regarding the known differences between genders and with aging in what concerns to metabolic syndrome prevalence in other populations and the linked
comorbidities that can be important to tackle also in the scenario of the Korean population. This more extensive discussion would give more soundness and interest to the readers worldwide.

Author Response

(The authors gave the same response as above.)

Round 2

Reviewer 1 Report

Dear authors,

In the methodology, in line 95 is section 2.3 and in 99 also, they must update this numbering and the following.

Thank you for making these changes.

Best regards.